# Justified Trust in AI Fairness Assessment using Existing Metadata Entities

**Alpay Sabuncuoglu**
The Alan Turing Institute

**Carsten Maple**
The Alan Turing Institute

## Abstract

AI is becoming increasingly complex, opaque and connected to systems without human oversight. As such, ensuring trust in these systems has become challenging, yet vital. Trust is a multifaceted concept which varies over time and context, and to support users in making decisions on what to trust, work has been recently developed in the trustworthiness of systems. This includes examination of the security, privacy, safety and fairness of a system; in this work we explore the fairness of AI systems. While mechanisms, such as formal verification, aim to guarantee properties such as fairness, their application in large-scale applications is rare due to cost and complexity issues. A major approach that is deployed in place of formal methods involves providing claims regarding the fairness of a system, with supporting evidence, to elicit justified trust in the system. Through continuous monitoring and transparent reporting of existing metadata with model experiment logs, organisations can provide reliable evidence for claims. This paper provides details of a new approach for evidence-based trust. We share our findings from a workshop with industry professionals and provide a practical example of how these concepts can be applied in a credit risk analysis system.

## 1 Introduction

Trust is the confidence, or belief, that one party meets another's reliability, integrity, and intentions in the value of protecting and maintaining the relationship Cook (2001). Building a mutual trust means developing a consistent and dependable relationship where both parties feel secure OECD (2017). Trust is linked to a variety of components including transparency, consistency, reliability, and credibility Lee & See (2004). When developing and deploying a technology for the common use, a trust should be established between the tech-deploying organisation and end-users. The organisation should take proactive approaches to prevent potential harms and demonstrate accountability in case these harms occur.

Establishing trust requires a comprehensive approach that prioritises transparency and accountability Williams et al. (2022). Jacovi et al. suggests establishing contracts to establish the trust with a clear commitment to make the model behaviour patterns available to the user by clear documentation, explanation, and analysis reports Jacovi et al. (2021). Accordingly, stakeholders involved in AI development are advised to design their risk management processes to routinely assess and monitor the trustworthiness of their systems Polemi et al. (2024). Evidence collection is a key component of this process, since it provides stakeholders with reliable information to support their decisions and actions. In the context of AI fairness, evidence collection involves responding to risks caused by different kinds of bias including statistical, cognitive, and social et al. (2021).

Evidence may be based on formal methods. In this case, a proof is provided showing that a machine learning model satisfies certain fairness properties, such as individual or counterfactual fairness Tumlin et al. (2024); Borca-Tasciuc et al. (2022); Yadav et al. (2024). This process aims to provide guarantees of equitable behaviour across all possible inputs, utilising techniques such as reachability analysis to confirm fairness.

However, fairness is not often asserted using formal verification, in contrast to other safety characteristics such as robustness, security and privacy. This is because a) fairness is a more subjective concept that is difficult to formalise; b) technical experts might fail to address the adequate societal implications of the system; c) the complexity of the fairness concept might require a more interdisciplinary approach; and d) the lack of a universal definition of fairness.

There has been limited success in establishing formal verification methods for fairness, and these usually on verify specific aspects of fairness, in specific contexts. In practice, researchers and auditors rather use a combination of assessment methodologies, such as testing, auditing, and red-teaming, to ensure that AI systems are fair. Demonstrating this type of reliable evidence can elicit "justified trust" in system design DSIT (2024). Justified trust refers to the level of trust that AI system's safety and compliance is based on clearly communicating reliable evidence to stakeholders. Building this type of trust for fairness requires a step by step evaluation of data, model, and interaction components in the system design and development process. For example, in the data collection phase, historical bias in the data can be identified by testing statistical disparity between privileged and unprivileged groups identified by protected characteristics. We can mitigate this harmful bias by using techniques like re-weighting. The experiment result is an evidence to reduce the risk of discrimination. However, other kind of biases can occur in model development, validation, and deployment stages. As such, continuous monitoring is required to support justified trust.

In this paper, we demonstrate maintaining continuous monitoring throughout the development lifecycle using existing metadata formats effectively. We align our metadata management and monitoring approach with existing risk management strategies to support seamless and effective adoption in industry setting. We also share our findings from a workshop with industry professionals and provide a practical example of how these concepts can be applied in a credit risk analysis system.

## 2 Fairness-related Metadata in the ML Lifecycle

Throughout the development of ML models, developers and other stakeholders use various documentation formats to enhance reproducibility and communicate the details of artefacts with both internal and external stakeholders. We refer to such documentation, designed to enhance clarity, accountability, and trust in ML systems, as *transparency artefacts*. We can use these transparency artefacts as justified evidence to verify the team took the required actions and catalogued evidence for the given arguments. The key requirement for this kind of evidence is they must be grounded in measurable and reproducible outcomes such as benchmarking results, auditing trails, or red teaming reports. In this section, we selected one documentation format for each main entity in the development process (data, model, and use case) to demonstrate the fairness-related fields.

### 2.1 Recording Data Metadata

Data cards provide detailed documentation of datasets used in training and evaluating ML models. They include information about data sources, preprocessing steps, limitations, and known biases, offering critical transparency about the data shaping model behaviour. This transparency is a valuable resource for identifying potential biases, evaluating model fairness, and understanding the broader sociotechnical challenges associated with the dataset Pushkarna et al. (2022).

> **Fairness-related metadata:** Google's Data Card specification mandates details such as funding sources, data subjects, representation balances of potential sensitive characteristics, dataset collection process, geographies involved in both collection and labelling processes, intended use cases, and potential unintended outcomes.

## 2.2    RECORDING MODEL METADATA

Model cards provide essential information about an ML model's intended use, limitations, and performance. They typically include details like model purpose, dataset used, evaluation metrics, ethical considerations, and any limitations or potential biases. To illustrate, Google's model card specification[1] includes nine main sections Mitchell et al. (2019): **(1)** Model details such as basic model information, **(2)** Intended use with the cases that were envisioned during development, **(3)** Factors could include demographic or phenotypic groups, environmental conditions, technical attributes, **(4)** Metrics chosen to reflect potential real world impacts of the model, **(5)** Evaluation data that was used for the quantitative analyses in the card, **(6)** Training data (when possible) to understand the distribution over various factors, **(7)** Quantitative analyses, **(8)** Ethical considerations, and **(9)** Caveats and recommendations. These information help users and other stakeholders to understand where and how a model should (and shouldn't) be applied.

> **Fairness-related metadata:** Entities of "Data" group can reveal potential representation bias and understand the sensitive characteristics. "Performance metrics of quantitative analysis" can include sensitive-group based analysis. If it is filled properly, provided confidence intervals show the statistical significance of these results. "Considerations" can support the next phase of requirement planning in an iterative development process. "License information" and its "SPDX" (System Package Data Exchange) link can reveal SBOM (software bill of materials) data for their development codebase. An SBOM document lists all the components that forms a software, including: Dependencies like libraries and API calls, and their versions and licenses. Particular versions of development libraries can introduce some measurement and evaluation bias throughout the development process.

## 2.3    RECORDING USE CASE METADATA

Based on the commonly used Unified Modelling Language (UML) in software engineering, Hupont et al. proposed use case cards as a standardised methodology to define intended purposes and operational uses of an AI system Hupont et al. (2023). A use case card consists of two main parts: (a) a canvas for visual modeling, and (b) a table for written descriptions. The canvas includes actors, use cases, and relationships, while the table provides additional details such as intended purpose, product type, safety component status, application areas, and other relevant information. Figure 1 illustrates a a use case card created for a financial sentiment analysis system, that we conducted a heuristic walkthrough in Section 3.

> **Fairness-related metadata:** This use case card format is designed in accordance with the EU AI Act, ensuring alignment with its requirements. The format's fields, such as direct links to open issues, provide valuable insights into fairness challenges at both the system and organizational levels.

## 2.4    RECORDING FAIRNESS-RELATED EXPERIMENT METADATA

Although, model, data and use case cards store some fairness-related metadata, they are not designed to address potential fairness recording needs. We created a fairness recording template to support development teams documenting key details related to the experimental setup, data characteristics, model specifications, and fairness evaluation metrics. The metadata begins with general information, providing a comprehensive overview of the experimental context. Figure 2 illustrates the metadata entities tracked in model, data, and fairness logs.

---

[1]https://modelcards.withgoogle.com/about

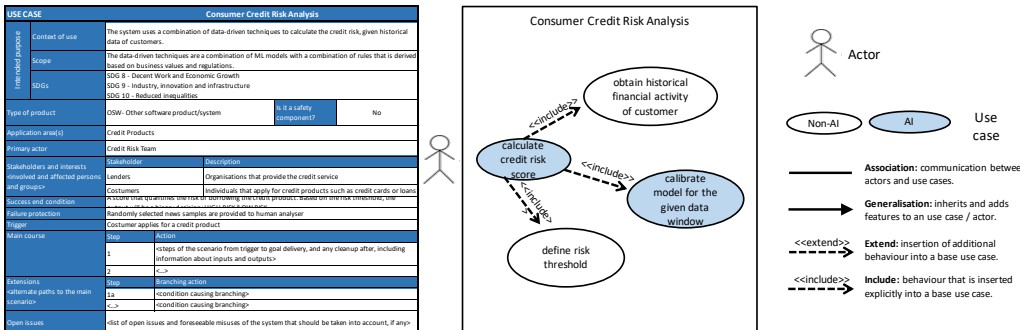

Figure 1: A simplified example use case card for a consumer credit risk calculation system. The template is sourced from Hupont et al. (2023).

The "data" section captures the characteristics of the dataset, such as "sample" details and profiles of key "variables". It explicitly identifies "sensitive_characteristics" (e.g., race, gender) that may influence fairness outcomes, alongside the categorisation of "nominal" and "continuous" features. The "model" section records the ML model's "name", ensuring traceability to the specific algorithm or architecture used. In the "sample_data" subsection, performance metrics such as true positives ("tps"), false positives ("fps"), true negatives ("tns"), and false negatives ("fns") are recorded, providing a foundation for quantitative analysis of model behaviour.

Finally, the "bias_metrics" section is pivotal in evaluating fairness. It organises metrics by "groups", where each "group_name" corresponds to a demographic or attribute category (e.g., age group, gender). Within each group, metrics are detailed with attributes such as their "name", "description", "value", and corresponding "thresholds". Parameters such as whether a higher metric value is preferred ("bigger_is_better") and additional "notes" or "subgroup (sg)" parameters further enrich the analysis. By facilitating granular tracking and assessment of fairness across groups, this template provides a robust mechanism for identifying and addressing bias, ultimately fostering ethical and responsible ML development.

We created a toolkit to support developers quickly integrate fairness-related metadata management into their codebase (open source but link removed for anonymity). While developers keep using their current experiment tracking tools (e.g. wandb, Neptune, mlflow), they can integrate toolkit's functionalities to record fairness-related data in a standardised and interoperable way. The toolkit includes templates and schemas for metadata storage, as well as utility files for initiating, populating and syncing metadata.

## 2.5 INSIGHTS FROM INDUSTRY PRACTITIONERS

To explore industry applicability, we organised a workshop with 11 professionals from diverse sectors, all with technical backgrounds but at different career stages. Participants included technical product managers, machine learning engineers, data scientists, R&D engineers, directors, and managers. While this paper does not focus on the workshop itself, the participants' feedback shaped our tool development and research agenda.

During the workshop, participants mapped potential biases in their use cases and identified key entities to monitor throughout the ML lifecycle. Figure 2 illustrates the results of this mapping exercise with highlighted bias types and corresponding metadata entities. Participant agreed that monitoring these entities can enable development teams to proactively address biases, ensuring AI systems remain fair and accountable. Appendix A provides a detailed overview of the workshop and participant profiles.

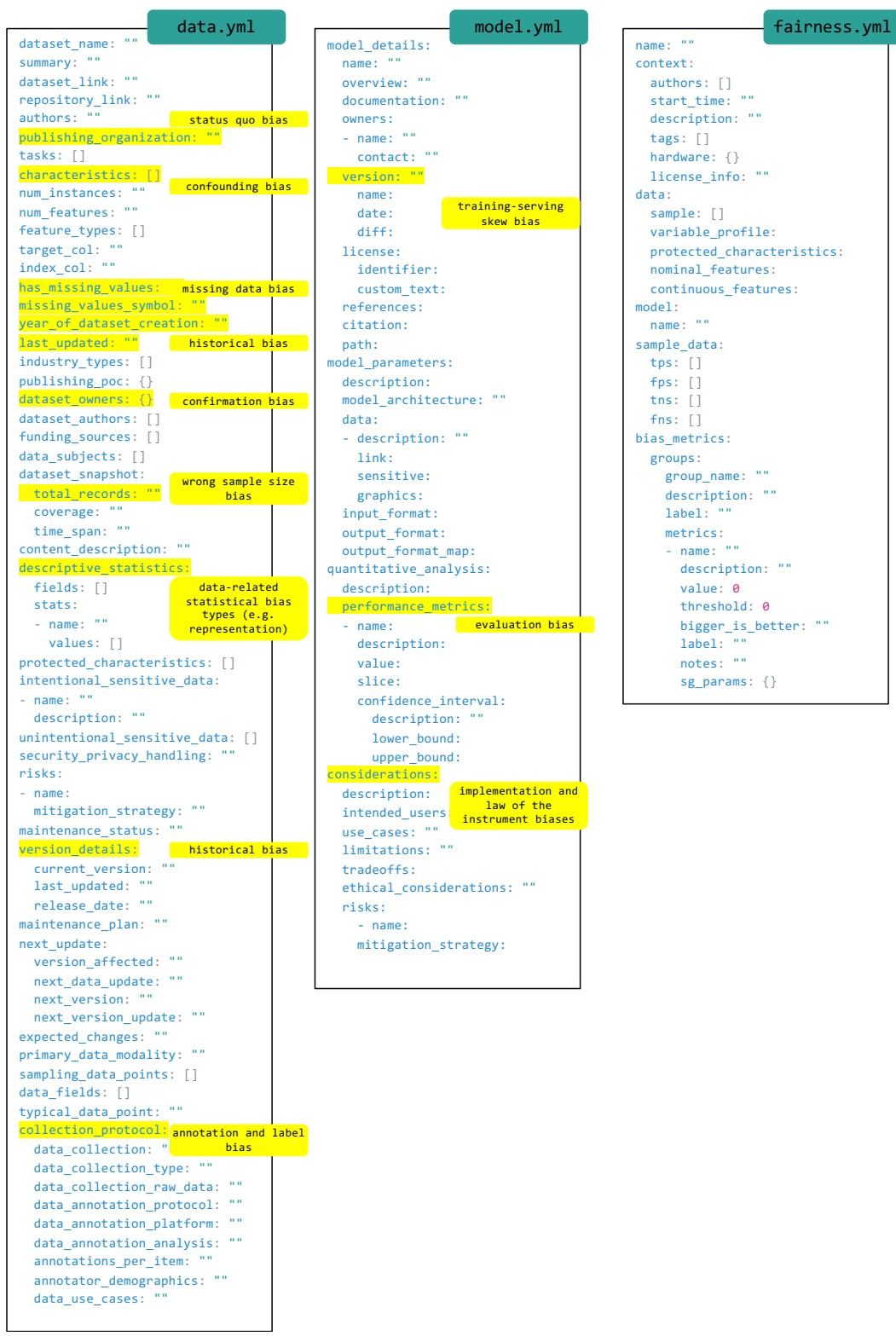

Figure 2: The list of metadata entities used in the model, data, and fairness experiment logs.

## 3 PRACTICAL DEMONSTRATION OF THE FLOW: FAIRNESS METADATA OF A CREDIT RISK ANALYSIS PROCESS

This section provides a step-by-step practical example of a fairness assessment for an ML-enabled creditworthiness assessment model in the UK consumer credit market. The use case assumes that the application is developed in the UK, hence the requirements are based on the UK regulations. We utilised open-source data, model, and libraries to share the complete pipeline open-source. Throughout this section, we demonstrate which metadata entities are updated and how our fairness metadata monitoring library utilised in this pipeline.

**Regulatory Guidance:** As a customer-facing service, the system must comply with the UK's Equality Act 2010, which prohibits discrimination based on protected characteristics such as sex, ethnicity, and age. It must also adhere to Financial Conduct Authority (FCA) regulations, which require fair treatment of customers FCA (2017). Compliance entails ensuring that credit decisions are free from bias against specific groups. Failure to meet these requirements could result in regulatory fines, loss of license, and reputational damage.

> **Update use case documentation:** Review regulatory compliance requirements and update the risk register accordingly. Include the relevant regulations and guidelines in the use case card documentation.

**Dataset:** In this exploratory use case, we use open-source models and datasets to ensure reproducibility and transparent documentation of our steps. Specifically, we utilise the German Credit Data Repository (1994), one of the most commonly used datasets in fairness research Fabris et al. (2022). This dataset consists of 1,000 individuals, each represented by 20 attributes, to classify them as good or bad credit risks. It includes both categorical and numerical features, such as the status of existing checking accounts, loan duration, credit history, purpose, and credit amount.

> **Update dataset documentation:** This dataset is available on UCI Machine Learning Repository. Include the dataset source and description in the data card documentation. Check fairness-related metadata in the existing data card and update it if necessary. Developers can use our metadata management library to automate this process:
>
> ```
> german_credit_data = fetch_ucirepo(id=144)
> pretty_metadata = pretty_uci_metadata(german_credit_data.metadata)
> for key, value in pretty_metadata.items():
> faidlog.add_data_entry(key=key, entry=value)
> ```
>
> The above code snippet fetches the German Credit Data, converts the metadata into an interoperable format, and attaches it to the existing data log.

**Protected characteristics accross the dataset:** The dataset includes "age" and "personal_status" features that can be classified as protected characteristics. The "personal_status" feature combines sex and marital status. In this use case, we focus on disparities between male and female groups. The dataset's sex distribution is 69% male and 31% female.

> **Update dataset documentation:** Include the protected characteristics in the data card documentation as well as the fairness experiment log. Check the representation balance of these characteristics in the dataset and update the data card's "descriptive statistics" field.

**Fairness Notions:** In creditworthiness assessment applications, researchers often use "equal opportunity" fairness notion. Hardt et al. (2016). This notion ensures that individuals with similar qualifications or circumstances receive similar outcomes, regardless of protected chracteristics-related factors. Statistical

parity difference (SPD) measures fairness by comparing the selection rates of different groups. It is calculated as the difference between the probability of a positive outcome for a priviliged group and a non-priviliged group. SPD can be used as one of the metrics to assess whether individuals with similar qualifications have equal chances of receiving a positive outcome, regardless of their group membership. A low SPD is one of the indicators to suggests fairness, but Equal Opportunity may still require checking disparities among individuals, hence requires utilising multiple metrics.

> **Update fairness experiment log:** Include the selected fairness notions and their definitions in the fairness log. In this use case, we can start by documenting statistical parity to reveal historical and representation biases. The positive outcome for the male group is 0.73, while positive outcome for the female group is 0.63. This indicates a -0.108 statistical parity difference in approval rates. You can report it in our standarised fairness metric schema:
>
> ```
> name: "statistical parity difference"
> description: "Statistical parity difference is the difference in approval
>     rates between two groups."
> value: -0.108
> threshold: 0 \#indicates perfect fairness, select this value based on
>     your usecase, regulatory requirements, and statistical significance.
> bigger\_is\_better: false
> label: "statistical\_parity\_difference"
> notes: ""
> sg\_params: \{\} \# we don't have any additonal subgroups in this case
> ```

**Model development:** The target feature is a binary value, "credit," which is a quantitative representation of a consumer is a "good" or "bad" candidate to give credit product. Let's develop different models to predict the creditworthiness of individuals. In this use case, we use a set of scikit-learn models (logistic regression, decision tree, random forest, gradient boosting, and support-vector) with default hyperparameters. We evaluate the models using accuracy, precision, recall, and F1 score.

> **Update model card:** Include the data source, model type, and evaluation metrics in the model card documentation. Check the fairness-related metadata in the existing model card and update it if necessary. Developers can use our metadata management library to automate this process:
>
> ```
> from faid.logging import ModelCard
> m = ModelCard()
> m.add_quantitative_analysis_metric({
>     'name': 'ROC AUC Score',
>     'description': 'ROC AUC Score is the area under the ROC curve',
>     'value': '0.878',
>     ... # other fields
> })
> ```

**Model Evaluation:** We evaluate the models using accuracy, precision, recall, and F1 score. We also assess the models' fairness using statistical parity, equalised odds, and calibration metrics. We compare the models' performance across different demographic groups to identify potential biases.

> Monitor the results of fairness-aware algorithms, such as re-weighting or adversarial debiasing, and define decision thresholds for fairness to have a pass/fail mechanism in build tests. Compare the performance of the model across different subgroups using the chosen metrics.

**Bias Mitigation:** If the model shows significant disparities in approval rates, error rates, or calibration, we can use bias mitigation techniques to address these issues. We can retrain the model with a balanced dataset, adjust the decision threshold, or use fairness-aware algorithms to reduce bias.

Analysis of these results and updating the risk score:

- Approval Rate: Disparity between males (75%) and females (65%), and between White (72%) and Minority (60%) groups. Indicates potential bias in the model's thresholds or features.
- Error Rates: Higher false negatives for females and minority groups suggest they are more likely to be incorrectly denied credit.
- Calibration: Higher calibration errors for minority groups indicate credit scores may not be equally predictive of repayment likelihood for these groups.

**Automated tests before deployment:** Automated tests is traditionally used to ensure quality of a system before deployment. In this use case, we can create automated tests to ensure that fairness metrics meet predefined thresholds before deploying the model. Since the fairness log already includes the metrics, values, and confidence intervals, we can use these values for defining a PASS/FAIL kind of test. Then, we can use this information to automatically inform the risk register and update risk scores to prioritise interventions and allocate resources to address fairness disparities.

```
from faid.scan import test_model_metadata_values
test_model_metadata_values()
```

The above code snippet can be used to test the model metadata values against predefined thresholds. It automatically collects fairness-related entities and turn them into PASS/FAIL tests based on the predefined thresholds.

**Risk Quantification Considerations:** The risk score is a measure of the potential impact of a risk event on the project. It is calculated by multiplying the probability of the risk event occurring by the impact of the risk event. The risk score is used to prioritise risks and allocate resources to address them. In this use case, the risk score can be updated based on the results of the fairness evaluation. If the model shows significant disparities in approval rates, error rates, or calibration, the risk score should be updated to reflect the potential impact of these biases on the project.

**Update risk register:** Our metadata management library comes with a minimal risk register implementation to sync GitHub issues with the fairness considerations. Using the existing Github templates and risk registers, development teams can open new issues to share it with other stakeholders in the project management process.
Update the risk register with the results of the fairness evaluation. Include the potential impact of fairness disparities on the project and the likelihood of these disparities occurring. Update the risk score to reflect the potential impact of these biases on the project.

## 4 CONCLUDING REMARKS

As, we elaborated throughout the use case, existing model logs and transparency artefacts can support this continuous monitoring process by automatically enabling sharing experiment data and integrating pass/fail mechanisms in build process. Several types of metadata and log files can store the information from a fairness analysis:

- Use case cards: Record use case considerations and compliance requirements based on organisational and environmental aspects such as hard laws, regulatory guidance, business goals, etc.
- Model cards: Keep updated versions of model cards for each retraining or major modification. Compare fairness metrics across versions to detect degradation or improvements. Share model cards with stakeholders to ensure transparency and accountability.
- Data sheets: Track changes in dataset composition over time to ensure consistent group representation. Use select data sheet entities to continuously monitor and verify that new datasets align with fairness goals.
- Fairness experiment Logs: Use logs to identify which configurations or datasets contribute to fairness disparities. Automate alerts when fairness metrics fall below a threshold during experimentation. Compare logs over time to track the impact of fairness-oriented interventions. Update bias reports at each stage to document changes. Use historical logs to identify recurring fairness issues. Integrate audit logs with automated pipelines for periodic re-evaluation.

This paper focus on the fairness dimension of trust in AI systems, but the same principles can be applied to other dimensions such as security, privacy, and robustness. By aligning existing risk management frameworks with continuous monitoring of ML development logs, organisations can build trust in their AI systems and ensure that they are fair and accountable. This approach can help organisations establish a culture of transparency and accountability, fostering trust among stakeholders and ensuring that AI systems are developed and deployed responsibly.

Deploying an industry-scale application is challenging, requiring thorough evaluation at both the system level (full-stack architectures, data and model versioning, etc.) and the organisational level (licenses, communities, team skills, etc.). A key takeaway from our workshop was that identifying biases and linking them to development-level entities helped professionals build a holistic understanding of sociotechnical concerns in system fairness.

However, automating this process is complex due to variations in architectures and version control mechanisms for data and models. In this rapidly evolving field, where libraries and architectures change frequently, achieving complete and robust automation remains difficult. Moreover, full automation may not be ideal, as developers could rely solely on automated checklists, neglecting manual assessments and introducing automation bias into the system.

Ultimately, design decisions hinge on the desired *level of trust* in the system. Greater automation places more trust in the technology, while increased manual oversight relies on human judgment. Striking the right balance is essential for establishing justified trust. By aligning existing risk management frameworks with continuous monitoring of ML development logs, organisations can foster confidence in their AI systems while ensuring fairness and accountability.

ACKNOWLEDGMENTS

This work is a part of Proactive Monitoring of AI Fairness research, which is supported by Innovate UK [Project No: 10108523]. This work was also supported, in whole or in part, by the Gates Foundation [INV-057591]. Under the grant conditions of the Foundation, a Creative Commons Attribution 4.0 Generic License has already been assigned to the Author Accepted Manuscript.

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

## A  APPENDIX: WORKSHOP

We brought together 11 participants with diverse technical expertise. All were professionals with technical backgrounds but at different career stages, including technical product managers, machine learning engineers, data scientists, R&D professionals, as well as directors and managers.

### A.1  WORKSHOP FLOW

We began with an introduction to fairness in AI and the key characteristics of trustworthy AI since our participants came from the industry and had limited experience with fairness studies. Throughout the workshop, participants worked on a use case, either a real-world application or a relevant theoretical scenario.

- **Defining the Use Case** The first activity focused on defining a use case by identifying three key components: (1) ML Task: Classification, regression, or generation (2) Model Type: Shallow or deep learning (3) Potential Data Sources: Tabular, image, audio, text, etc.

- **Understanding Bias and Regulatory Context:** Next, we introduced protected characteristics and how current regulations address fairness in ML model development. Using insights from the recent International AI Safety Report, we demonstrated different bias types, including: Sampling bias, selection bias, historical bias, labeller bias, evaluation bias, and feedback loop bias.

- **Identifying Bias with Bias Type Cards:** Participants then used Bias Type Cards et al. (2021). Each card featured the bias name and a brief description as well as a more detailed explanation with speculative prompts to help identify the bias. Participants selected the most relevant biases for their use cases and mapped them to specific ML lifecycle stages where developers and stakeholders should be particularly vigilant.

- **Metadata Management and Bias Monitoring:** We introduced our workflow, focusing on metadata management and technical considerations. Participants explored how existing metadata could be leveraged to monitor potential biases across different ML stages.

- **Building Monitoring Groups:** In the final interactive activity, participants selected relevant data, model components, and fairness log entities to form "monitoring groups." These groups were designed to proactively track values and detect potential bias-related risks.

- **Workshop Conclusion:** At the end of the session, each group presented their work and provided feedback to one another.

## A.2 PARTICIPANT PROFILE

**Their current use and sectoral perspective of AI:** They reported involvement in a diverse range of ML/AI development efforts, including healthcare applications, information processing, broader consultation on model evaluation and governance, and infrastructure and data management. We also asked the question to understand their sectoral perspectives on AI development and their motivation in participating in the workshop. They highlighted the following *short-term trends:* The adoption of LLMs for their existing business applications. Enhanced efficiency through AI-driven automation in workflows. Integration of LLMs into everyday tasks, including search, information retrieval, and translation. Improved cost-benefit analysis of machine learning (ML) versus traditional approaches.And, some of the *long-term trends* they identified are: Evolution toward agentic AI systems capable of autonomously executing user-defined tasks. Increasing AI independence in handling complex, non-human-performable tasks. Seamless ML integration with minimal additional burden on developers. Development of standardised and trusted metrics for LLM fairness. Growing emphasis on trust and transparency in AI implementation.

These trends indicate that they have already received requests from their clients or companies to integrate AI into their current products/services. But, they are still struggling in the assessment of these products using reliable and robust metrics.

**Their challenges:** Workshop participants identified several major challenges in AI development, including: Difficulty in keeping up with fast-moving innovations and evolving AI capabilities. Challenges in sourcing, cleaning, and maintaining high-quality data for AI models. Ensuring comprehensive assessment of LLMs and AI systems throughout their lifecycle. Balancing speed of iteration from prototype to production while maintaining robust evaluation. High costs associated with training, processing, storage, and clinical trials. Selecting appropriate vendors and integrating AI systems effectively.

**The main barriers for them to the responsible adoption and use of AI systems:** Difficulty in understanding AI metrics, selecting safe and ethical solutions, and developing an AI strategy that ensures responsible use. Lack of support from management and concerns about integrating AI responsibly within business operations. Ambiguity around legal and ethical boundaries, particularly regarding data privacy and compliance. Poor data governance structures, limited access to key data (e.g., protected characteristics), and over-reliance on third-party systems with opaque decision-making ("black boxes"). High costs of AI infrastructure, limited innovation capacity, and difficulty in attracting AI talent. Startups and smaller firms face pressure to iterate quickly, often without sufficient time to assess the full impact of AI decisions.

When we talk about fairness as a trustworthiness component, the conversation is structured around transparency. They emphasised the need for an interpretable mechanism with clear decision-making processes.

**Top ask of policy-makers:** Regulations should consider the entire AI value chain, including environmental impacts. AI policies should be concise, easy to understand, and available in summarisable formats. Provide worked examples and gold-standard deliverables to help organisations comply effectively.

