# OpenReview forum: "Justified Trust in AI Fairness Assessment using Existing Metadata Entities"
_ICLR.cc/2025/Workshop/BuildingTrust — BuildingTrust_

### Official Review · Reviewer_KSLP · 2025-03-01

**Rating:** 8
**Confidence:** 2

**Review:**

The paper tackles an important problem, and justifies their framework through a real use case. This paper seems like very relevant to the BuildingTrust workshop.

Some relevant papers to cite regarding the definition of trust:
1. John D Lee and Katrina A See. 2004. Trust in Automation: Designing for Appropriate Reliance. Human factors, 46.
2. Alon Jacovi, Ana Marasovic, Tim Miller, and Yoav Goldberg. 2021. Formalizing Trust in Artificial Intelligence: Prerequisites, Causes and Goals of Human Trust in AI. In ACM Conference on Fairness, Accountability, and Transparency (FAccT).

---

### Official Review · Reviewer_aLxr · 2025-03-02
**Justified Trust in AI Fairness Assessment Using Existing Metadata Entities**

**Rating:** 6
**Confidence:** 2

**Review:**

This paper proposes leveraging model logs and transparency artifacts (data cards, model cards, fairness logs) to build evidence-based trust in AI fairness claims. The methodology is demonstrated through a credit risk assessment case study and informed by industry practitioner feedback.

Strengths:

- Addresses practical, evidence-based trust rather than abstract fairness principles with emphasis on continuous monitoring
- Structured approach using existing metadata artifacts aligned with industry standards and regulations
- Incorporates real-world practitioner feedback from industry workshops on implementation challenges
- Concrete credit risk case study demonstrating regulatory alignment (UK Equality Act, FCA rules)
- Useful discussion on automating fairness monitoring through metadata logging

Weaknesses and Suggestions:

- Lacks empirical validation of effectiveness beyond documentation practices
- Limited case study scope focused only on credit risk assessment
- Insufficient clarity on how this approach complements or replaces active fairness interventions
- Workshop findings presented without systematic analysis of practitioner feedback
- Does not critically examine effectiveness of existing regulations or address gaps
- Limited novelty beyond organizing existing best practices
- Unclear connection between metadata tracking and actionable fairness improvements
- Ambiguous balance between automation and necessary human oversight

I rate this paper 6 because it is a practical approach to AI fairness monitoring aligned with industry practices, but requires additional empirical validation and broader application testing to demonstrate effectiveness beyond documentation.

---

### Official Review · Reviewer_WBy7 · 2025-03-02

**Rating:** 6
**Confidence:** 2

**Review:**

## Summary
The paper proposes a standardized format for reporting information about an ML system's fairness. This is done by augmenting currently used meta-data (e.g. the model card and the data card) with more explicit fairness data, as well as a new log for experiments and results around fairness. The paper also demonstrates an application of this format of meta-data in for a credit risk analysis application, and proposes some simple heuristic based checks for ensuring fairness from the parsed meta-data files.

## Strengths
* The paper addresses an important issue for ensuring trust in ML models.
* The core of the paper's proposal can be added onto existing meta-data files (i.e. data, model cards) which are already released in ML workflows
* The case-study clarifies the contributions and the applications of the proposed system.

## Weaknesses
I would like to preface this section by stating that I do not have enough experience as an industrial ML practitioner, or an HCI/fairness researcher to critique this paper authoritatively.
* The paper does not seem to add much over and above existing systems around model releases in my opinion. The paper also does not clarify what incentive corporations would have to release additional meta-data around fairness (aside from the case-study of EU regulations)
* Some of the proposals are a bit impractical for larger ML systems (particularly LLMs), where training data is not revealed (or audited properly), and model cards do not contain much information about the training procedure either. An idealized case study in this setting could help guide future model releases greatly.

---

### Decision · Program_Chairs · 2025-03-04

Accept